# Study on the Interaction between Wheel Polygon and Rail Corrugation in High-Speed Railways

**DOI:** 10.3390/ma15248765

**Published:** 2022-12-08

**Authors:** Xiaotian Xu, Xiaolu Cui, Jia Xu, Xiaoxia Wen, Zongchao Yang

**Affiliations:** 1School of Mechanotronics & Vehicle Engineering, Chongqing Jiaotong University, Chongqing 400074, China; 2China Railway Materials Operation and Maintenance Technology Co., Ltd., Beijing 100036, China

**Keywords:** high-speed railway, wheel polygon, rail corrugation, friction coupling vibration, competition mechanism

## Abstract

The wheel polygonization and rail corrugation are typical wheel–rail periodic wear problems, which seriously affect the safe operation of high-speed railways. In the present paper, the interaction between the wheel polygon and the rail corrugation in the long-slope section of high-speed railways is mainly studied based on theory of friction coupling vibration. Firstly, the simulation model of the wheel–rail contact model is established, as well as the polygonal wear of the wheel and the corrugated wear of the rail. Then, the stability analyses of the wheel–rail system with periodic wear are studied, in which the four working conditions of smooth rail–smooth wheel, polygonal wheel–smooth rail, smooth wheel–corrugated rail and polygonal wheel–corrugated rail are compared. Finally, the competition mechanisms between the wheel polygon and rail corrugation under different parameters are discussed, including the wheel–rail friction coefficient and the depth of periodic wear of the wheel–rail system. The numerical results show that both the periodic wear of the wheel and rail with certain relevance will increase the friction coupling vibration of the wheel–rail system, which may aggravate the subsequent relevant wheel polygonal and rail corrugation wear. With the increase of the friction coefficient between wheel and rail, as well as the depth of the wheel polygon and rail corrugation, the vibration trend of the friction coupling vibration of the wheel–rail system increases gradually. Moreover, the proportion of the wheel polygon’s influence on the friction coupling vibration of the wheel–rail system is greater than that of rail corrugation.

## 1. Introduction

The high-speed railway has the advantages of high safety, high stability and high comfort. The operating mileage of China’s high-speed railways has exceeded 40,000 km, ranking first in the world. As the core technology of high-speed railways, the wheel–rail relationship reflects the dynamic interaction between the wheel and the rail [1]. The traction, braking and driving safety of the train are closely related to the wheel–rail contact. Especially with the development of high-speed railways, the wheel–rail problem has become more complicated [2]. The wheel–rail periodic wear is a hot issue in the field of the wheel–rail relationship. The periodic wear of the wheel is represented by the polygonal wear on the wheel tread, while the periodic wear of the rail is represented by the corrugated wear on the rail surface, which are referred to as the wheel polygon and rail corrugation, as shown in Figure 1 [3]. It can be found that both the wheel tread and rail surface have periodic wear with alternating wave crest and wave valley, and these two problems in high-speed railways have not been effectively solved so far. Among them, in the study of the wheel polygonization, the current mainstream viewpoint is that the generation and development of the wheel polygon is related to the vibration of the vehicle–track system, including the vibration of the bogie system, the vibration of the wheel axle system and the vibration of the wheel–rail system. First of all, from the perspective of the vibration of the bogie system, Wu et al. [4] believed that the modal coupling vibration between the bogie frame and wheelset would lead to 23-order polygonal wear of the wheels on the high-speed train. Liu et al. [5] presented that the coupling vibration of the wheelset bending vibration and the motor vertical vibration causes 9-order polygonal wear of the wheel. Then, from the perspective of the vibration of the wheel axle system, Morys et al. [6] proposed that the first-order bending vibration of the wheelset leads to the rapid development of the initial third-order wheel polygonization. Frohling et al. [7] considered that the first-order bending vibration of the wheelset is the fundamental reason for the wear of 20-order wheel polygonization. Next, from the perspective of the wheel–rail system resonance, Chen et al. [3] put forward the hypothesis that the friction self-excited vibration of the wheel–rail system causes the wheel polygonization of the high-speed train. Tao et al. [8] found that P2 resonance of the wheel–rail system is the cause of the 5–8-order wheel polygonization. Cai et al. [9] found that the third-order bending vibration of the rail is the genesis of higher-order polygonal wheel wear. Then, in the study of the rail corrugation, the generation mechanism of rail corrugation is mainly studied around the inherent vibration characteristics of the wheel–rail system and the feedback vibration characteristics of rail surface irregularities. On the one hand, from the perspective of the inherent vibration of the wheel–rail system, Correa et al. [10] studied the influences of four different high-speed track structures on the development of rail corrugation. Cui et al. [11] researched the impacts of different vehicle types on rail corrugation in high-speed railways, and explored the dynamic causes of rail corrugation. Yu et al. [12] established a wheel–rail unsteady rolling contact model for high-speed trains, and studied the impact of wheel–rail contact on the rail corrugation considering the geometric nonlinear relationship between wheel and rail. On the other hand, from the perspective of the feedback vibration of rail surface irregularities, Braghin et al. [13,14] combined the dynamic model, Kalker vibration theory and wear model to build a non-uniform prediction model of rail wear, and analyzed the influence of random excitation of rail irregularity on rail corrugation. Zhao et al. [15] analyzed the short-wave corrugation in high-speed railways based on the theory of wavelength fixation mechanism and proposed a theory to explain the gradual stabilization of rail corrugation.

Based on the above studies, it is found that the wheel polygon and rail corrugation are similar in terms of wear characteristics, and some of the formation mechanisms are also very close, but the carriers that occur are different. Therefore, there is a common question, does the wheel polygon affect the rail corrugation, and does the rail corrugation affect the wheel polygon? Is there any correlation between the two problems? However, most research only focuses on a single problem at present, but the interaction between wheel polygonization and rail corrugation is less studied [16,17]. The competition mechanism between the wheel polygon and rail corrugation needs to be further studied. Section 2 describes the methodology of the present paper, including the numerical simulation model of the wheel–rail system, as well as the polygonal wear of the wheel and the corrugated wear of the rail, and the theoretical method of friction coupling vibration analysis. Section 3 presents the stability analyses of the wheel–rail system with periodic wear, in which the four working conditions of smooth wheel–smooth rail, polygonal wheel–smooth rail, smooth wheel–corrugated rail and polygonal wheel–smooth rail are compared. Section 4 further discusses the competition mechanisms between wheel polygon and rail corrugation under different parameters, including the wheel–rail friction coefficient and depth of periodic wear of the wheel–rail system. Finally, the conclusions are summarized.

## 2. Simulation Model and Theoretical Method

### 2.1. Finite Element Model of Wheel–Rail Contact

Most lines of high-speed railways are straight lines or curved lines with large radii (radius greater than 1000), and there are relatively few curves with small radii. The present paper mainly studies the interaction between the wheel polygon and the rail corrugation under the condition of a straight line with long-slope section. This is because the long-slope section is the high-incidence interval of rail corrugation [18]. It should be explained that typically the curves with a small radius are more vulnerable to corrugation, but curves with a radius less than 450 m do not occur in high-speed railways. Rail corrugation mainly occurs on the straight lines where trains brake or accelerate. That was the reason why long-slope sections are more vulnerable to corrugation.

Figure 2 shows the wheel–rail contact model in the straight line [3]. Through the suspension forces (F_SVL_ and F_SVR_) at the left and right ends of the wheelset, the wheelset contacts with the rails. The normal contact forces (N_L_ and N_R_) and the tangential forces (F_L_ and F_R_) are generated between the wheel and rail, and the corresponding contact angles are δ_L_ and δ_R_. Ignoring the influence of the serpentine motion of the wheelset, the wheel–rail contact states on the left and right sides of the straight line are basically the same [19]. Moreover, the track is mainly connected with the under-rail structure by fasteners, which can be simulated by spring-damper units. The vertical stiffness and damping are K_RV_ and C_RV_, respectively, and the lateral stiffness and damping are K_RL_ and C_RL_, respectively.

According to the above wheel–rail contact model, the relevant finite element model of wheel–rail contact model is built, as shown in Figure 3. The simulation model mainly includes the wheelset, two rails and a series of fastener systems. The rolling circle diameter of the wheels in the wheelset is 920 mm with the LMA wheel tread. The wheel–rail contact is applied as a surface-to-surface contact scheme with a penalty contact algorithm, and the friction coefficient is set as 0.3 [20]. The rail length is set at 25 m with fixed constraints on both rail ends. The fastener systems are mainly modeled by vertical, lateral and longitudinal 3 × 7 spring-damper elements, in which the vertical stiffness and damping of the fastener are set as K_V_ = 50 MN/m and C_V_ = 30 kNs/m, and the lateral and longitudinal stiffness and damping are set as K_L_ = 28 MN/m and C_L_ = 20 kNs/m. The corresponding material properties are taken primarily from Ref [21]. The elastic modulus of the wheelset is 210,000 MPa, the density is 7800 kg/m^3^, and the damping ratio is 0.3. Additionally, the elastic modulus of the rail is 210,000 MPa, the density is 7800 kg/m^3^ and the damping ratio is 0.3.

Moreover, considering the periodic wear characteristics of wheel and rail in this contact model, referring to the theoretical models of wheel polygonization and rail corrugation in Section 2.2, the polygonal wear of the wheel tread and the corrugated wear of the rail surface are respectively built by modifying the surface nodes of the wheel or rail, as shown in Figure 3. It should be noted that, since the wheel polygonization occurs at the left and right wheels of the wheelset, and the rail corrugation occurs at the left and right sides of the straight line [22,23], both sides of the wheelset and the rails will be considered in the subsequent construction of the wheel polygonal wear and the rail corrugated wear. Through field research and literature reviews of high-speed railways, the typical order of the wheel polygon is about 18–22-order, and the common wavelength of rail corrugation is about 120–160 mm [24].

### 2.2. Theoretical Models of Wheel Polygonization and Rail Corrugation

In the construction of periodic wear on the wheel tread and rail surface in the finite element model, its geometric characteristics should be considered. This section briefly introduces the theoretical models of wheel polygonization and rail corrugation.

Firstly, for the simulation of polygonal wear of the wheel, the “rose line” equation is assumed to be the ideal equation [25]. The numerical expression of the “rose line” equation of the *n*-stage polygonal wear in polar coordinates is:(1)$$p_{n}(\theta )=0.5A(1-\cos(n\theta )),0\leq \theta \leq 2\pi$$
where *A* is the envelope radius of the polygonal wear and *θ* is the angle. By subtracting the “rose line” equation of the *n*-stage polygon from the contour line of the smooth wheel, it is the ideal contour equation of the polygonal wheel after wear, that is:(2)$$p(\theta )=R_{0}-p_{n}(\theta )=R_{0}-0.5A(1-\cos(n\theta )),0\leq \theta \leq 2\pi$$
where *R*_0_ is the radius of the smooth wheel tread profile.

Taking the 20-order wheel polygon as an example, the image of the “20-leaf rose line” is shown in Figure 4a, and the 20-order worn polygon wheel is shown in Figure 4b. Moreover, according to Equation (2), the node coordinates of the wheel tread in the finite element model of wheel–rail contact model shown in Figure 4 are corrected. The relevant polygonal wear surface of the wheel can be established.

Then, for the simulation of the corrugated wear of the rail, the sin function is assumed to be an ideal equation [26]. The numerical expression of the *n*-stage corrugated wear of the rail is:(3)$$Z_{0}(l)=\frac{1}{2}D\left(1-\cos\frac{2\pi l}{L}\right)(0\leq l\leq nL)$$
where *Z*_0_(*l*) is the corrugated irregularity, *D* is corrugation depth, *L* is the corrugation wavelength, *l* is the length of corrugated wear and *n* is the order of corrugated wear.

Taking the corrugated wear with the wavelength of 140 mm as an example, the image of the corrugated wear is shown in Figure 5. Moreover, according to Equation (3), the node coordinates of the rail surface in the finite element model of wheel–rail contact shown in Figure 5 are corrected. The relevant corrugated wear surface of the rail can be established.

### 2.3. Theoretical Method of Friction Coupling Vibration Analysis

As one of the mechanisms of studying wheel polygonization and rail corrugation, the friction coupling vibration theory can reasonably explain the formation and development of the wheel polygon and rail corrugation to a certain extent [11,19]. There are certain assumptions in the friction coupling vibration theory, that is, the creep force between the wheel and rail tends to be saturated. It is assumed that the saturated creep force between wheel and rail will induce the friction coupling vibration of wheel–rail system, thus inducing and intensifying the formation and development of wheel polygonization and rail corrugation [3,25]. This is usually suitable for small radius curve sections and long-slope braking or traction sections, which are also high-incidence sections for rail corrugations [17]. This paper also takes the long-slope section of high-speed railways as the research object, and assumes that the creep force between the wheel and rail tends to be saturated. Therefore, the friction coupling vibration of wheel–rail system can be used as the key indicator to determine the severity of the wheel polygon and the rail corrugation. 

The complex eigenvalue analysis method is a common method to study the friction coupling vibration characteristics of the wheel–rail system [27,28]. The theoretical derivation process is summarized as follows. Firstly, it is assumed the mass matrix is [***M***], the stiffness matrix is [***K***] and the damper matrix is [***C***]. The corresponding motion equation of the whole wheel–rail system is: (4)$$[M]\ddot{x}+[C]\dot{x}+[K]x=q,$$
where $\ddot{x}$, $\dot{x}$ and *x* are the node acceleration, the node velocity and the node displacement of the system. *q* is the wheel–rail friction force, which is assumed as an equation of the displacement times the contact stiffness and friction coefficient:(5)$$q=\mu [K_{f}]x,$$
where [***K****_f_*] is the matrix of localized contact stiffness, *μ* is the friction coefficient. Thus, its motion equation is rewritten as:(6)$$[M]\ddot{x}+[C]\dot{x}+[[K]-\mu [K_{f}]]x=0,$$
where [***M***], [***C***] and [[***K***] − *μ*[***K****_f_*]] are asymmetric matrixes. The characteristic equation is represented as: (7)$$(\lambda^{2}[M]+\lambda [C]+[[K]-\mu [K_{f}]])\varphi =0,$$
where *λ* and *φ* are the eigenvalue and eigenvector of the system, respectively.

Since it is troublesome to settle the eigenvalue problem of the asymmetric matrix in Equation (7), the subspace projection method is used to simplify the above equation [28], that is:(8)$$(\lambda^{2}[M^{*}]+\lambda [C^{*}]+[K^{*}])\varphi_{p}=0,$$
where [***M***^∗^], [***C***^∗^] and [***K***^∗^] are the simplified symmetric mass, damping and stiffness matrixes, respectfully. *φ*_p_ is the eigenvector of the system. 

Furthermore, Equation (8) is settled by the standard QZ method, and the general analytic solution is obtained as below:(9)$$x(t)=\sum_{i=1}^{n}\varphi_{i}e^{\lambda_{i}t}=\sum_{i=1}^{n}\varphi_{i}e^{(\alpha_{i}+j\omega_{i})t},$$
where *λ_i_* = *α_i_* + *jω_i_*. *α_i_* is the real part of the complex eigenvalue and *ω_i_* is the imaginary part of the eigenvalue. The positive real part means that the system may experience unsteady vibrations. The larger the positive real part is, the more unsteady the system is. Therefore, the real part of the complex eigenvalue can be used as a key parameter to evaluate the friction coupling vibration of the wheel–rail system, so as to judge the severity of the wheel polygonization and rail corrugation.

## 3. Comparative Study of Different Wheel–Rail Contact Conditions

To explore the relationship between the rail corrugation and wheel polygonization, the stability analyses of the wheel–rail system with periodic wear are studied, in which the four working conditions of smooth rail and smooth wheel, polygonal wheel and smooth rail, smooth wheel and corrugated rail, and polygonal wheel and corrugated rail are compared, as shown in Figure 6. The friction coupling vibration characteristics of the system under different wheel–rail contact states are compared, so as to explore the stability of the system. Based on the viewpoint of the friction coupling vibration, it is believed that the friction coupling vibration of the wheel–rail system may cause the unstable vibration of the wheel–rail system, thus inducing or aggravating the formation and development of the wheel polygon or the rail corrugation [29,30].

The typical order of the wheel polygon is about 18–22-order, and the common wavelength of rail corrugation is about 120–160 mm in high-speed railways [24]. The 20-order wheel polygon with a depth of 0.05, and the rail corrugation with the wavelength of 140 mm and a depth of 0.04 are selected in the present study. According to the complex eigenvalue analysis, the friction coupling vibration characteristics of the system under the four wheel–rail contact conditions are shown in Figure 7, Figure 8, Figure 9 and Figure 10. Figure 7 shows the distribution of the real part of eigenvalues and modal situations of the smooth wheel–smooth rail system. It can be found that there are two very close unstable vibration frequencies of the system in this contact state, which are 350.22 Hz and 349.84 Hz, respectively. The corresponding unstable vibration modes of the system are on the left and right wheels, respectively. It needs to be explained here that the two unstable vibrations are very close in frequency, and it can be considered that the unstable vibration of the wheel–rail system mainly occurs on both sides of the wheelset at the frequency of about 350 Hz. The unstable vibration frequencies and modes correspond to the rail corrugation with the wavelength of 140 mm and 20-order wheel polygon, which means that the there is indeed some relevance between the rail corrugation and the 20-order wheel polygon. In addition, by amplifying the unstable vibration modes, it is found that unstable vibrations also occur on the left and right sides of rails. This means that under the smooth wheel–smooth rail contact condition, both the wheel polygon and rail corrugation may occur under the action of the wheel–rail friction coupling vibration, and the possibility of wheel polygonization is higher than that of rail corrugation in high-speed railways [31]. Moreover, Figure 8 shows the distribution of real parts of eigenvalues and modal situations of the polygonal wheel–smooth rail system. Figure 9 shows the distribution of real parts of eigenvalues and modal situations of the smooth wheel–corrugated rail system. Figure 10 shows the distribution of real parts of eigenvalues and modal situations of the polygonal wheel–corrugated rail system. It can be found that the friction coupling vibrations of the wheel–rail system occur at about 350 Hz, and appear on both the left and right wheels and rails. The main distinguishing features between the above four working conditions are that the values of the real part of the eigenvalue are different.

Moreover, Table 1 shows the competition between the friction coupling vibration characteristics of the system under different wheel–rail contact states. The real part of the eigenvalue is an important indicator to judge the stability trend of the system [29]. It can be found that in conditions 2 and 3, when there is periodic wear on one side of the wheel and the rail, the real part of the eigenvalue of the system shows an increasing trend relating to the condition 1. The increasing trends of the real part of the eigenvalue of the system in conditions 2 and 3 are almost the same, in which the effect of the polygonal wheel on the stability of the entire system is slightly greater than that of rail corrugation. Furthermore, when there is periodic wear on both the wheel and the rail in condition 4, the real part of the eigenvalue of the system is the largest. It means that when the wheel polygon and rail corrugation with certain relevance exist together, the friction coupling vibration of the wheel–rail system is the strongest, which means that the system is the most unstable. Moreover, the discussion of the interaction between the rail corrugation with other wavelengths and other order wheel polygonization can be found in Ref. [32]. When the wheel polygon and rail corrugation are without relevance, they will be further discussed in the future.

In conclusion, the above comparisons can be concluded that whether the periodic wear of the wheel or the rail may increase the friction coupling vibration of the wheel–rail system, which aggravates the subsequent relevant order wheel polygon and rail corrugation with relevant wavelength. It is most serious when the wheel polygon and rail corrugation with certain relevance exist together.

## 4. Parametric Analyses

To further study the competition mechanism of the mutual influence of wheel polygonization and rail corrugation under different parameters, the parametric analyses are carried out in this section. The effects of the wheel–rail friction coefficient, the depth of corrugated wear and polygonal wear on the friction coupling vibration of the wheel–rail system are studied.

### 4.1. Wheel–Rail Friction Coefficient

As an important factor affecting the wheel–rail wear, the wheel–rail friction coefficient is affected by many aspects, such as climates and regions [33]. The distribution of eight vertical and eight horizontal high-speed railways in China leads to large differences in the wheel–rail friction coefficient in different climates and regions. Taking the four wheel–rail contact conditions studied above as the research objects, with the change of the wheel–rail friction coefficient, the variations of the real parts of eigenvalues under different wheel–rail contact conditions are shown in Figure 11. The variation range of the wheel–rail friction coefficient is set as 0.1 to 0.5 [34,35]. Only the maximum real parts of the eigenvalue under different working conditions are compared. From the overall trend, it can be found that the real part of the eigenvalue of the system increases with the increase of the wheel–rail friction coefficient under the four working conditions. Moreover, comparing the four working conditions, it can be found that the overall trend is still that the real part of the eigenvalues of the polygonal wheel–corrugated rail system is the largest, followed by the polygonal wheel–smooth rail system, then the smooth wheel–corrugated rail system, and finally the smooth wheel–smooth rail system. Then, by paying attention to local details shown in Figure 11b, it can be found that when the friction coefficient changes from 0.1 to 0.2, no matter the action of wheel polygonization or rail corrugation, the growth rate of the real part under each working condition is not obvious. When the friction coefficient changes from 0.2 to 0.3, the growth rate of the real part increases significantly, especially in the polygonal wheel–corrugated rail system. When the friction coefficient is larger than 0.3, the growth rate of the real part decreases significantly. Furthermore, especially comparing the polygonal wheel–smooth rail system and the smooth wheel–corrugated rail system, when the wheel–rail friction coefficient is less than 0.3, the real parts of these two working conditions are very close. However, after the wheel–rail friction coefficient is larger than 0.3, the growth rate of the real part of the polygonal wheel–smooth rail system is significantly greater, which means that the influence of the polygonal wheel on the stability of the entire wheel–rail system is more obvious than the rail corrugation.

To summarize, it can be concluded that with the increase of wheel–rail friction coefficient, the friction coupling vibration of wheel–rail system will be intensified, which aggravates the development of the wheel polygon and rail corrugation. Moreover, comparing the competition mechanism of wheel polygonization and rail corrugation on the friction coupling vibration of wheel–rail system, the influence of the wheel polygon on the stability of the entire system is higher than that of the rail corrugation, especially when the wheel–rail friction coefficient is greater than 0.3. 

### 4.2. Depth of Periodic Wear

With the increase of running mileage of high-speed trains, the depth of wheel polygon and rail corrugation is also deepening. The periodic wear depth of the wheel or rail has a significant effect on the friction coupling vibration of the wheel–rail system [32]. Regarding the influences of the depth variations of wheel polygonization and rail corrugation on the friction coupling vibration characteristics of the wheel–rail system, the changes of the real part of the eigenvalue of the wheel–rail system are shown in Figure 11, including the depth variation of wheel polygon, depth variation of rail corrugation and depth variations of both wheel polygon and rail corrugation. From Figure 12a, the depth of the wheel polygon varies from 0 to 0.1 mm at 0.025 mm intervals. From Figure 12b, the depth of the rail corrugation varies from 0 to 0.08 mm at 0.02 mm intervals. From Figure 12c, the depth of the wheel polygon and rail corrugation varies together at the above variation ranges. It can be found that the whether the depth of the wheel polygon or the depth of the rail corrugation increases, the real part of the eigenvalue of the wheel–rail system shows an obvious increasing trend. Comparing Figure 12a with Figure 12b, with the increase of depth in the wheel polygon, the slopes of the fitting curve of the real part of the eigenvalue are 17.78 and 15.84, respectively, while with the increase of depth of rail corrugation, the slopes of the fitting curve of the real part of the eigenvalue are 13.32 and 10.02, respectively. It can be concluded that the change of depth of the wheel polygon has a more obvious influence on the stability of the entire wheel–rail system than the change of depth of rail corrugation, which proves again that the proportion of wheel polygon’s influence on the friction coupling vibration of wheel–rail system is greater than that of rail corrugation. Figure 12c shows the increase trend of real part of the eigenvalue of the wheel–rail system is the most obvious when the depth of the wheel polygon and rail corrugation increases together. It can be preliminarily concluded that with the increase of the depth of wheel polygon and rail corrugation, the vibration trend of the friction coupling vibration of the wheel–rail system increases gradually. When the wear depth of wheel and rail increases together, the instability trend of the system is most obvious.

## 5. Conclusions

Based on the theory of friction coupling vibration, the interaction between the wheel polygon and rail corrugation of high-speed trains is studied in the present paper. The long ramp section of high-speed railways is taken as the main research interval, and the simulation models of the wheel–rail system with periodic wear including wheel polygonization and rail corrugation are constructed, respectively. Then, the friction coupling vibration characteristics of the wheel–rail system under four typical working conditions are compared. Finally, the competition mechanism between the wheel polygon and rail corrugation is further explored through parametric analyses. The relevant conclusions are summarized as follows.

(1) Both the periodic wear of the wheel and the rail will increase the friction coupling vibration of the wheel–rail system, which may aggravate the subsequent relevant wheel polygonization and rail corrugation. It is most serious when the wheel polygon and rail corrugation with certain relevance exist together.

(2) With the increase of the friction coefficient between wheel and rail, the friction coupling vibration of the wheel–rail system will be intensified, which aggravates the development of the wheel polygon and rail corrugation.

(3) With the increase of the depth of the wheel polygon and rail corrugation, the vibration trend of the friction coupling vibration of the wheel–rail system increases gradually. When the wear depth of wheel and rail increases together, the instability trend of the system is the most obvious.

(4) The proportion of the wheel polygon’s influence on the friction coupling vibration of the wheel–rail system is greater than that of rail corrugation.

## Figures and Tables

**Figure 1 materials-15-08765-f001:**
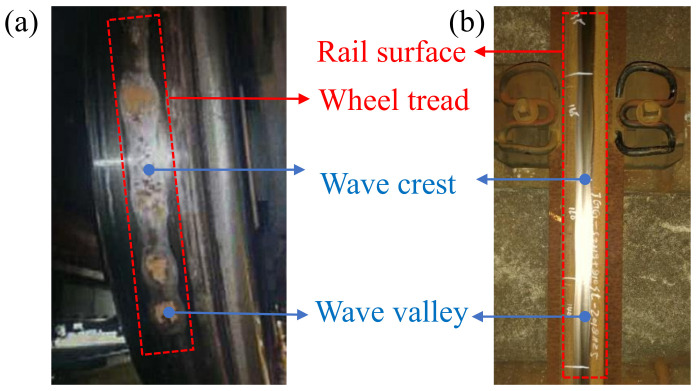
Wheel–rail periodic wear: (**a**) wheel polygon [3] and (**b**) rail corrugation.

**Figure 2 materials-15-08765-f002:**
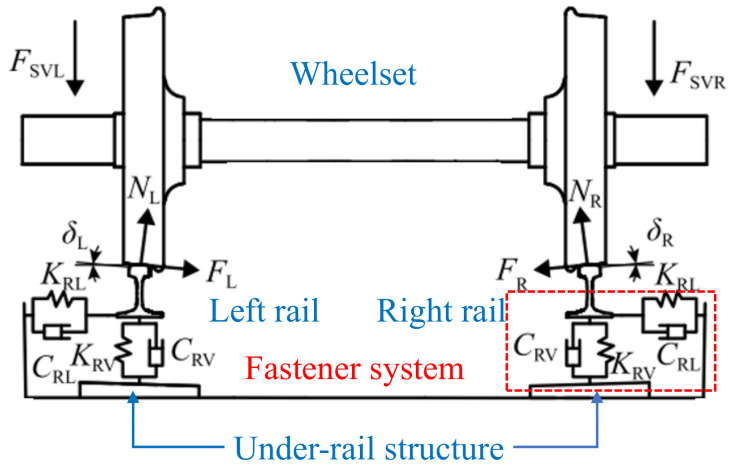
Wheel–rail contact model in straight line.

**Figure 3 materials-15-08765-f003:**
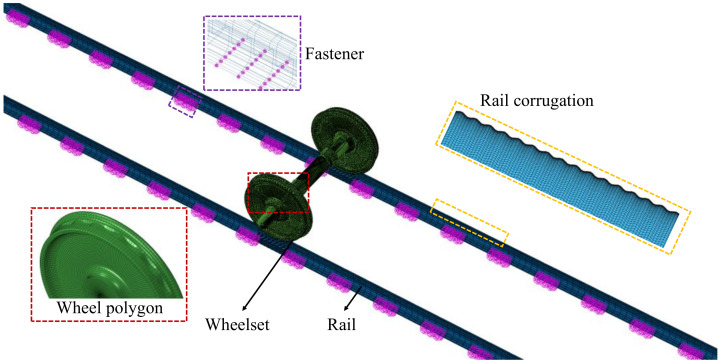
Finite element model of wheel–rail contact model.

**Figure 4 materials-15-08765-f004:**
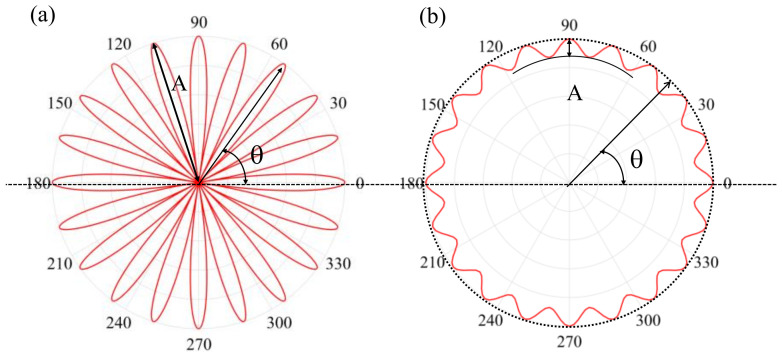
Design of 20-order wheel polygon: (**a**) 20-leaf rose line and (**b**) 20-order wheel polygon.

**Figure 5 materials-15-08765-f005:**
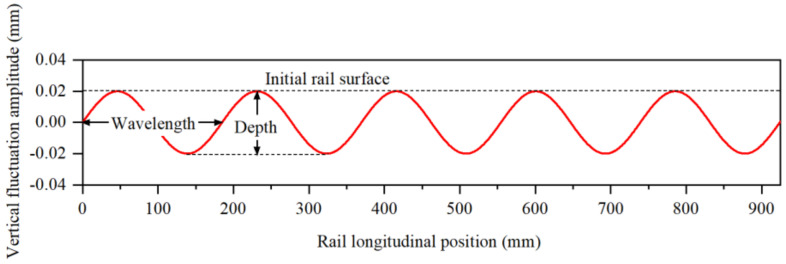
Design of rail corrugation with the wavelength of 140 mm.

**Figure 6 materials-15-08765-f006:**
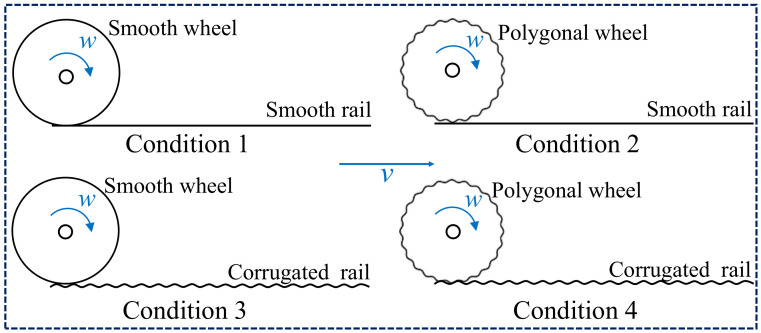
Different wheel–rail contact conditions.

**Figure 7 materials-15-08765-f007:**
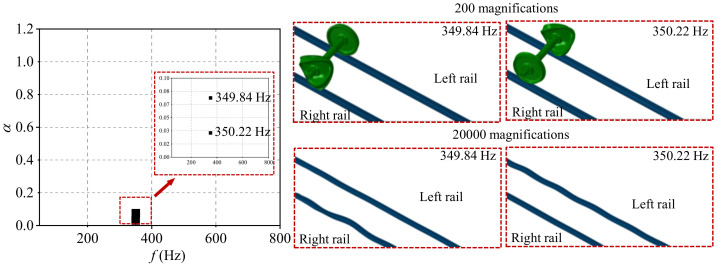
Distribution of real parts of eigenvalues and modal situations of the smooth wheel–smooth rail system.

**Figure 8 materials-15-08765-f008:**
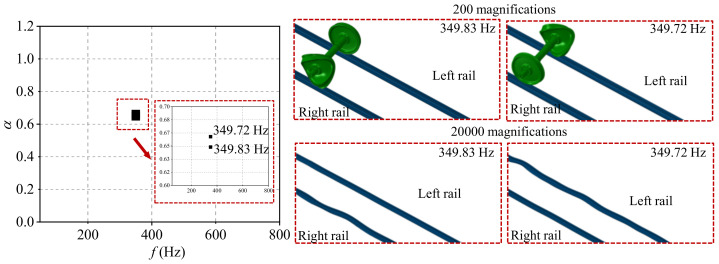
Distribution of real parts of eigenvalues and modal situations of the polygonal wheel–smooth rail system.

**Figure 9 materials-15-08765-f009:**
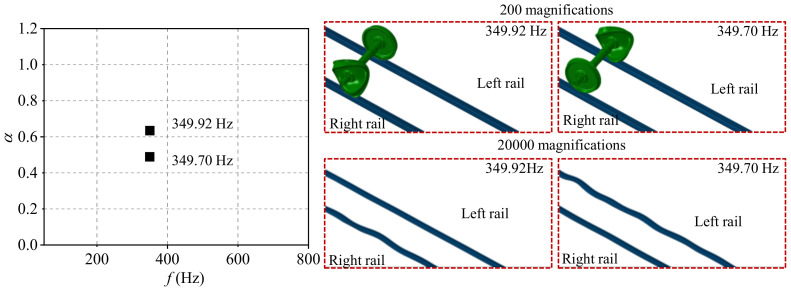
Distribution of real parts of eigenvalues and modal situations of the smooth wheel–corrugated rail system.

**Figure 10 materials-15-08765-f010:**
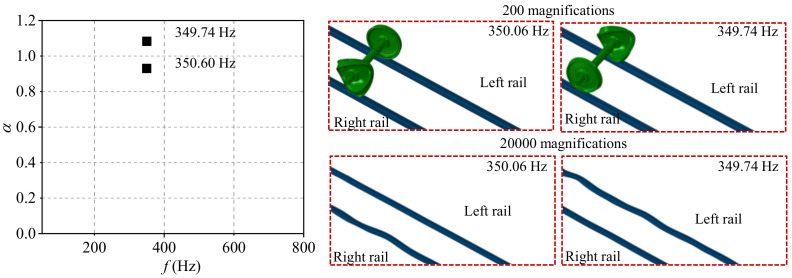
Distribution of real parts of eigenvalues and modal situations of the polygonal wheel–corrugated rail system.

**Figure 11 materials-15-08765-f011:**
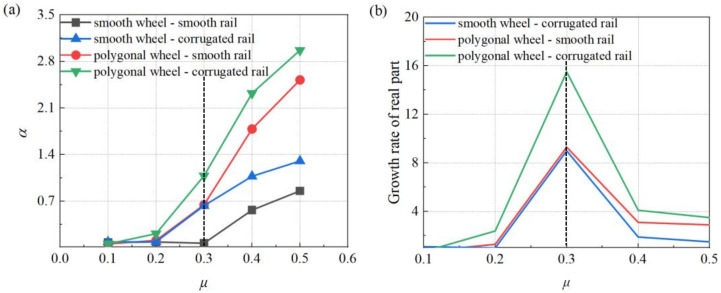
Effect of the wheel–rail friction coefficient on the stability of the system: (**a**) distribution of the real part; (**b**) growth rate of the real part.

**Figure 12 materials-15-08765-f012:**
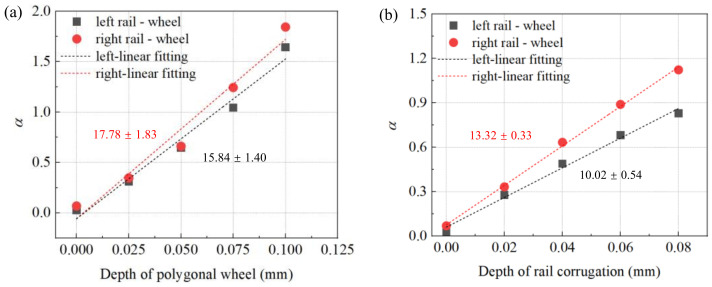
Effect of the depth of periodic wear on the stability of the wheel–rail system: (**a**) depth variation of polygonal wheel; (**b**) depth variation of rail corrugation; (**c**) depth variations of both wheel polygon and rail corrugation.

**Table 1 materials-15-08765-t001:** Friction coupling vibration characteristics of the system under different wheel–rail contact states.

Condition	System	Real Part	Frequency	Location
1	smooth wheel–smooth rail	0.0307	349.84 Hz	Left
		0.0749	350.22 Hz	Right
2	polygonal wheel–smooth rail	0.6619	349.83 Hz	Left
		0.6487	349.72 Hz	Right
3	smooth wheel–corrugated rail	0.4892	349.92 Hz	Left
		0.6343	349.70 Hz	Right
4	polygonal wheel–corrugated rail	0.9303	350.06 Hz	Left
		1.0833	349.74 Hz	Right

## Data Availability

Not applicable.

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
