# Peer review of "Study on the Interaction between Wheel Polygon and Rail Corrugation in High-Speed Railways"

_materials, 2022, doi:10.3390/ma15248765_

Round 1

Reviewer 1 Report

In the study titled "Study on the interaction between wheel polygon and rail corrugation in high-speed railways", possible reinforcement and analysis in the case of contact of wheel and rail with different surfaces were tried to be analyzed theoretically by simulation.

However, information should be given about the possible material properties analyzed. Depending on the conditions presented in the study, especially the fatigue performance of these materials should also be analyzed. Finally, necessary statistical analyzes should be performed for the reliability of the study.

Reviewer 2 Report

The work is purely theoretical in nature and it completely lacks any comparison with real experimental data.

When calculating the resonant behavior of a wheelset, higher resonances are not considered and the mechanical properties of the axle and the vibrational interaction of the left and right wheels are not taken into account.

When writing equations (2-4), there is no reference to Figure 2 and it is not clear what the authors mean by the x coordinate in this equation.

In formula (2-3), x is meant to be the coordinate along the wheel's generatrix, as in Figure 5.

Accordingly, equations (2-6), (2-7) and (2-8) are written for some generalized masses and coordinates and it is absolutely not clear how the authors were able to force friction (equation (2-5)) enter into the differential equation (2-8) in the form of just the effective stiffness K*.

Apparently, the cited publications describe the method used by the authors to analyze the process of oscillatory behavior of a wheelset. However, the attention that the authors pay to the fact that there are two slightly different resonant frequencies for the left and right wheels, suggests that they do not understand the reason for the difference in the values issued by the calculation program.

In equation (2-6), the effect of friction on the behavior of the oscillatory system is attributed to the term describing the elastic reaction of the system. In further analysis, friction does not affect the resonant frequency in any way, but it changes the Q-factor of the system and the coefficient α describing the rate of increase of oscillations, equation (2-9).

Reviewer 3 Report

"There are few curved lines in high-speed railways, and most of the lines are straight lines." What does it means? Can you give me the example of HSR without curves? I agree that the tortuosity of HST is lower than conventional line. But this sentence is incorrect.

"This is because the long-slope section is the high-incidence interval of rail corrugation" -> Typically the curves with small radius are more vulnerable to corrugation. What was pointed out in [XXVIII]. But in the HSR lines, the curves with radius less than 450 m doesn't occurs. The second reason of the corrugation are the section of raiway line, where trais brake or accelerate. That was the reason, why long-slope sections are more vulnerable to corrugation.

Finite element model of wheel-rail contact: What type of railway construction was modelled? Ballast or ballastless? Both types showing different vulnerable to corrugation. What are the length beetween the rail joints?

Figure 1 - the rail wheel shown in the picture a) is worn to a degree that is to high in the normal exploatation. Or maybe it was taken intentionally with angle and light to highlight the problem. This wheel looks like the flat spot not a typical wheel poligon.

The numerical analisysy (pictures 7-10) showing that the variability of the vibration frequency is insignificant for the assumed  wheel poligon and corugation parameters. To rule out  an apparent dependency, it is worth to  analise wheel 18 order and corrugation with wavelength 120 mm, and 160 mm oraz something like that.

"In conclusion, the above comparisons can be concluded that whether the periodic wear of the wheel or the rail may increase the friction coupling vibration of the wheel-rail system, which aggravates the subsequent wheel polygon and rail corrugation. It is most serious when the wheel polygon and rail corrugation exist together." -> This conclusion is essential, that why you should check the apparent dependency, as it is mentioned above.

Can you indicate the conclusion, that homogenous motion (only HSR) is conductive to wear?

What was missing in work? The analysis not only the rail and wheel, bu the whole railroad. In the case of the rigid under-rail structure (which simulates concrete slab, which after all bend and transfer the low frequency vibrations  12-30 Hz up to 100Hz) and wery short model (25 m), high vibration frequencies could be expected.

Round 2

Reviewer 1 Report

Desired corrections and additions, the answers given are sufficient. It is appropriate to publish in this state.